# BroRL: Scaling Reinforcement Learning via Broadened Exploration

**Jian Hu** [1]  **Mingjie Liu** [1]  **Ximing Lu** [1]  **Fang Wu** [2]  **Zaid Harchaoui** [3]  **Shizhe Diao** [1]  **Yejin Choi** [1]
**Pavlo Molchanov** [1]  **Jun Yang** [1]  **Jan Kautz** [1]  **Yi Dong** [1]

## Abstract

Reinforcement Learning with Verifiable Rewards (RLVR) has emerged as a key ingredient for unlocking complex reasoning capabilities in large language models. Recent work ProRL (Liu et al., 2025a) has shown promise in scaling RL by increasing the number of training steps. However, performance plateaus after thousands of steps, with clear diminishing returns from allocating more computation to additional training. In this work, we investigate a complementary paradigm for scaling RL: **BroRL**—increasing the number of rollouts per example to hundreds to exhaustively **Bro**aden exploration, which yields continuous performance gains beyond the saturation point observed in ProRL when scaling the number of training steps. Our approach is motivated by a mass balance equation analysis allowing us to characterize the rate of change in probability mass for correct and incorrect tokens during the reinforcement process. We show that under a one-step RL assumption, sampled rollout tokens contribute to correct-mass expansion, while unsampled tokens outside rollouts may lead to gains or losses depending on their distribution and the net reward balance. Importantly, as the number of rollouts per example $N$ increases, the effect of unsampled terms diminishes, making overall correct-mass expansion more likely. To validate our theoretical analysis, we conduct simulations under more relaxed conditions and find that a sufficiently large rollout size $N$—corresponding to ample exploration—can increase the probability mass of correct tokens broadly, and in our simulator it increases all correct-token probabilities and eliminates knowledge shrinkage. Empirically, BroRL revives models saturated after 3K ProRL

training steps and demonstrates robust, continuous improvement, achieving strong results for the 1.5B model across diverse benchmarks. Notably, under the same training time, BroRL is both more data- and compute-efficient: large-$N$ rollouts reduce the number of filtered samples during dynamic sampling at the algorithmic level and nearly double generation throughput compared to ProRL in our hardware setup; this throughput increase is consistent with shifting generation from a more memory-bound regime toward a more compute-bound one.

## 1. Introduction

One of the central drivers behind the rapid advances in Large Language Models (LLMs) over the past a few years has been the discovery and application of *Scaling Laws*. Kaplan et al. (2020) showed that model performance follows predictable power-law improvements with respect to parameters, data, and compute. Building on this, Hoffmann et al. (2022) demonstrated that training is compute-optimal when model size and training tokens are scaled proportionally. These insights powered breakthroughs from GPT-3 to Claude/GPT-4 era, where scaling laws guided compute-optimal training of larger and more capable models.

More recently, Reinforcement Learning with Verifiable Rewards (RLVR) has brought new excitement to the field, unlocking complex reasoning in LLMs and fueling the rise of large reasoning models such as DeepSeek-R1 (Guo et al., 2025) and OpenAI-o1 (Jaech et al., 2024). Yet, *how to effectively scale the RLVR paradigm* remains an open question. Recent work ProRL (Liu et al., 2025a; Hu et al., 2025b) has demonstrated the potential of scaling RL by increasing the number of training steps. While this approach yields steady initial gains, performance plateaus after thousands of steps, with clear diminishing returns from allocating more computation to additional training.

In this work, we investigate a complementary dimension of the RL scaling law: **BroRL**—increasing the number of rollouts per example to hundreds to exhaustively **Bro**aden exploration. We position BroRL as a late-stage rollout-

---

[1]NVIDIA [2]Stanford University [3]University of Washington. Correspondence to: Yi Dong <yidong@nvidia.com>, Jan Kautz <jkautz@nvidia.com>.

*Proceedings of the 43$^{rd}$ International Conference on Machine Learning*, Seoul, South Korea. PMLR 306, 2026. Copyright 2026 by the author(s).

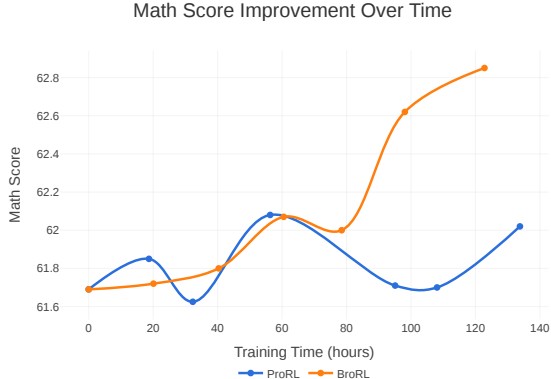

*Figure 1.* Empirical results demonstrate that BroRL ($N = 512$) continues to improve math performance, whereas ProRL ($N = 16$) reaches a plateau at the 3k-steps checkpoint and further degrades with prolonged training. The x-axis is **total wall-clock time**, including both rollout generation and gradient updates.

allocation recipe for saturated RLVR training, rather than a new optimizer or a replacement for efficient small-$N$ early training. Intuitively, our approach mirrors how humans tackle hard problems (e.g., four color theorem), making countless attempts over decades until a breakthrough emerges. Theoretically, our approach is motivated by a mass balance equation analysis. As shown in Figure 2, under the one-step RL assumption, the change in correct-token probability mass $\Delta Q_{\mathrm{pos}}$ consists of two parts. (1) The sampled portion contributes a non-negative gain by promoting sampled-correct tokens and demoting sampled-incorrect tokens. (2) The unsampled portion is conditional, potentially adding or removing mass depending on the batch distribution. Importantly, as the number of rollouts per prompt $N$ increases, the influence of the unsampled terms diminishes, and the overall effect tends toward $\Delta Q_{\mathrm{pos}} \geq 0$.

To verify our theoretical analysis, we conduct simulations with a TRPO-style linear surrogate objective. The results show that a sufficiently large rollout size $N$—corresponding to ample exploration—increases the probability mass of correct tokens and, in our simulator, eliminates knowledge shrinkage (Wu et al., 2025), i.e., worst-case probability reductions among correct tokens. This suggests that with enough exploration, RLVR can acquire new knowledge with reduced risk of forgetting in this controlled setting. Building on this foundation, we apply the BroRL recipe to scale RL training on real-world reasoning models. In particular, we continue training the ProRL model that plateaus after 3K steps and find that BroRL yields robust, continuous performance improvements, achieving strong results for the 1.5B model across diverse benchmarks.

Notably, under the same training time, BroRL is both more data- and compute-efficient: large-$N$ rollouts reduce the number of filtered samples during dynamic sampling at the

algorithmic level and nearly double throughput compared to ProRL in our hardware setup, consistent with shifting generation from a more memory-bound regime toward a more compute-bound one. BroRL highlights the central role of exploration in RL, revealing that the perceived limits of RLVR are sometimes artifacts of algorithmic design (e.g., insufficient rollouts) rather than the fundamental limits of RL itself, underscoring the necessity and promise of future algorithmic advances in RL.

**Conflict of Interest Disclosure.** J.H., M.L., X.L., S.D., Y.C., P.M., J.Y., J.K., and Y.D. are employed by NVIDIA, which developed and released the Nemotron-Research-Reasoning-Qwen-1.5B model used as the base model in our experiments.

## 2. Theoretical Analysis

We develop a theoretical analysis based on a mass balance argument, common in physics for mass and transfer analysis. Our analysis is performed in the logit domain, focusing on the partial mass of correct tokens (negative tokens, respectively). By a common abuse of language, we shall regularly use "probability" to refer to a logit [1].

**Notation.** We consider a vocabulary of size $V$, with logits $z \in \mathbb{R}^V$ and probabilities $p = \mathrm{softmax}(z)$. Let $\mathcal{P}$ and $\mathcal{N}$ denote the sets of correct and incorrect tokens in vocabulary $V$, respectively. $N$ rollout tokens are sampled, where each sampled token receives a binary reward $R_i \in \{R_c, R_w\}$ depending on whether it is correct or incorrect, while unsampled tokens are assigned $R_i = 0$. In the standard setting, the rewards satisfy $R_c \geq 0 \geq R_w$. Let $A \subseteq \mathcal{P}$ be the set of sampled correct tokens, $B \subseteq \mathcal{N}$ the set of sampled incorrect tokens, and $U$ the set of unsampled tokens.

Let the partial mass $P_{\mathrm{pos}}$ denote the total probability mass of the sampled correct tokens, and $P_{\mathrm{neg}}$ the total probability mass of the sampled incorrect tokens. Similarly, let $Q_{\mathrm{pos}}$ be the total probability mass of all correct tokens, and $Q_{\mathrm{neg}}$ the total probability mass of all incorrect tokens.

$$P_{\mathrm{pos}} = \sum_{i \in A} p_i, \qquad P_{\mathrm{neg}} = \sum_{i \in B} p_i,$$
$$Q_{\mathrm{pos}} = \sum_{i \in \mathcal{P}} p_i, \qquad Q_{\mathrm{neg}} = 1 - Q_{\mathrm{pos}}.$$

The corresponding second moments, which measure how

---

[1]This language is unrelated to the confidence we may assign to a logit and whether the model is statistically calibrated or not (Geng et al., 2023; Liu et al., 2025b).

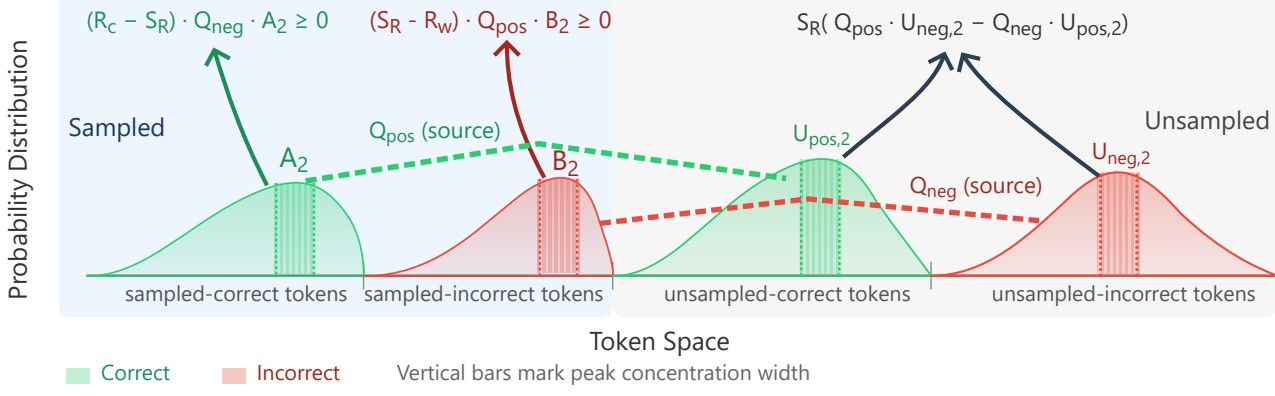

*Figure 2.* This illustration shows how a single RLVR update step alters the total probability mass $\Delta Q_{\text{pos}}$ for correct tokens, where the dashed guide lines labeled $Q_{\text{pos}}$ (green) and $Q_{\text{neg}}$ (red) connect the pooled probability assigned to the correct and incorrect token sets across sampled and unsampled regions. The change is composed of two parts: the Sampled portion (left) always produces a nonnegative gain by promoting "sampled-correct" tokens (concentration measured by $A_2$) and demoting "sampled-incorrect" tokens (concentration measured by $B_2$), thereby shifting probability from the $Q_{\text{neg}}$ pool to the $Q_{\text{pos}}$ pool. The unsampled part (right) is conditional: it can add or remove mass depending on the batch mood $S_R$ and whether unsampled incorrect probability is more concentrated than unsampled correct probability. As the number of samples per prompt $N$ grows, the unsampled concentration terms $U_{\text{pos},2}$ and $U_{\text{neg},2}$ shrink, so the net effect tends toward $\Delta Q_{\text{pos}} \geq 0$; the amount of mass moved scales with the pool sizes $Q_{\text{pos}}$ and $Q_{\text{neg}}$.

each partial mass is concentrated, are given by:

$$A_2 = \sum_{i \in A} p_i^2, \qquad B_2 = \sum_{i \in B} p_i^2,$$

$$U_{\text{pos},2} = \sum_{i \in U \cap \mathcal{P}} p_i^2,$$

$$U_{\text{neg},2} = \sum_{i \in U \cap \mathcal{N}} p_i^2.$$

Finally, define $S_R = \sum_{k \in A} R_c \, p_k + \sum_{k \in B} R_w \, p_k := R_c \, P_{\text{pos}} + R_w \, P_{\text{neg}}$ which represents the net contribution of sampled tokens, balancing the rewards from correct and incorrect tokens.

**Connection between** pass@$k$ **and** $Q_{\text{pos}}$  The quantity $Q_{\text{pos}}$ denotes the total probability mass assigned to correct tokens. For a single task, let $Q_{\text{pos}}(x) \in [0,1]$ denote the total probability mass assigned to correct tokens for input $x$. When drawing $k$ i.i.d. samples, the per-task success probability for input $x$ is

$$\text{pass@}k(x) = 1 - \big(1 - Q_{\text{pos}}(x)\big)^k.$$

This expression is strictly increasing in $Q_{\text{pos}}(x)$; thus, RLVR updates that increase the positive probability mass directly improve pass@$k$, and at a geometric rate. Taking the expectation over the task distribution yields

$$\mathbb{E}_x[\text{pass@}k(x)] := 1 - \mathbb{E}_x\big[(1 - Q_{\text{pos}}(x))^k\big]. \quad (1)$$

Moreover, if $Q_{\text{pos}}(x)$ increases pointwise (i.e., $Q'_{\text{pos}}(x) \geq Q_{\text{pos}}(x)$ for all $x$, with strict inequality on a set of positive measure), then both pass@$k(x)$ and its expectation increase strictly.

**One-step RLVR update.**  We perform our analysis under the simplifying assumption of a single step of RLVR, which allows us to obtain convenient analytical formulas. We model a single RLVR step as adjusting logits $z \in \mathbb{R}^V$ via a gradient update with rewards $\{R_c, R_w\}$ on sampled tokens. The update induces a first-order change in token probabilities $\Delta p$, which aggregates into a total correct-mass change

$$\Delta Q_{\text{pos}} = \sum_{i \in \mathcal{P}} \Delta p_i,$$

where $\mathcal{P}$ is the set of correct tokens. Then we show that the one-step change decomposes into an *sampled positive* term (always nonnegative) and an *unsampled coupling* term (which can be negative but vanishes as the rollout size $N$ grows). This decomposition allows us to uncover *distinct dynamics* in each term. In particular, the scaling of each of these terms with respect to $N$ leads us to identify the *rollout size* as a key quantity for trading off exploration breadth, prompt batch size, and update depth in experiments.

Formally:

**Theorem 1 (Sign of Correct-Mass Change).**

$$\Delta Q_{\text{pos}} := \frac{\eta}{N}\Big[(R_c - S_R)Q_{\text{neg}}A_2 \ + \ (S_R - R_w)Q_{\text{pos}}B_2$$
$$+ \ S_R\big(Q_{\text{pos}}U_{\text{neg},2} - Q_{\text{neg}}U_{\text{pos},2}\big)\Big],$$

*where* $A_2, B_2 \geq 0$, *and* $S_R \in [R_w, R_c]$, *which implies* $R_c - S_R \geq 0$ *and* $S_R - R_w \geq 0$. *Therefore, the first two terms are nonnegative, while the last term represents the coupling of unsampled masses.*

**Interpretation.** Three terms account for the change in the

probability mass of correct predictions, as illustrated in Figure 2.

The first term, $(R_c - S_R)Q_{\text{neg}}A_2$, arises from sampled-correct tokens. Each correct token has an advantage of $(R_c - S_R)$, meaning it is explicitly rewarded. Normalization redistributes this reward by taking mass from the incorrect pool (the $Q_{\text{neg}}$ share), and the size of the effect grows when those correct tokens are highly concentrated (large $A_2$). This term is always nonnegative: pushing up correct tokens can never reduce correct probability. This is a key feature of the sampled-correct tokens component of the reinforcement dynamics.

The second term, $(S_R - R_w)Q_{\text{pos}}B_2$, arises from sampled-incorrect tokens. These have an effective (negative) advantage of $(R_w - S_R) \leq 0$, so their probabilities are pushed down. Normalization then routes the freed-up mass to the correct pool in proportion to its size ($Q_{\text{pos}}$), and the effect is stronger when the incorrect samples were concentrated (large $B_2$). Again, this is always nonnegative: pushing down incorrect tokens leaves more probability for correct ones.

The third term, $S_R\big(Q_{\text{pos}}U_{\text{neg},2} - Q_{\text{neg}}U_{\text{pos},2}\big)$, comes from unsampled tokens, and unlike the first two, it can be positive or negative. Here the batch "mood" $S_R$ sets the direction: If $S_R > 0$ (a reward-positive batch), unsampled logits are nudged downward. This helps if unsampled incorrect mass is more concentrated (large $U_{\text{neg},2}$), but hurts if unsampled correct mass is more concentrated (large $U_{\text{pos},2}$). If $S_R < 0$ (a reward-negative batch), the signs flip: unsampled logits are nudged upward. This helps if unsampled correct tokens are more concentrated, but hurts if unsampled incorrect mass dominates.

Thus, the first two terms always contribute positively, while the third can either reinforce or oppose them depending on batch balance and how probability is distributed among unsampled tokens.

We draw several implications: (i) As the per-prompt sampling size $N$ grows, the unsampled terms $U_{\text{pos},2}, U_{\text{neg},2}$ shrink, which makes $\Delta Q_{\text{pos}}$ increasingly likely to be nonnegative (and in the limit of full sampling, the coupling term vanishes). (ii) Even for small $N$, positivity holds under balanced batches ($S_R \approx 0$) or when unsampled mass is sufficiently small. (iii) Increasing per-prompt sampling size $N$ directly improves pass@k by enlarging the positive margin of $\Delta Q_{\text{pos}}$.

Since pass@k$(x)$ is monotone in $Q_{\text{pos}}(x)$, any step with $\Delta Q_{\text{pos}} > 0$ improves success probability. Larger $N$ strengthens this effect by reducing the contribution of the third (unsampled) term, which can be negative under certain conditions. Full derivations and proofs are in Appendix A.1.

**Expected decay of unsampled mass.** The coupling term in Theorem 1 depends on the unsampled second moments $U_{\text{pos},2}, U_{\text{neg},2}$. These shrink as the rollout size $N$ grows:

**Lemma 2.** *Let a token with probability $p$ be sampled independently in each of $N$ draws. The expected "unsampled second-moment" contribution of this token is*

$$\mathbb{E}[U_2(p)] = p^2(1-p)^N.$$

*Corollary* 3. For a collection of tokens with probabilities $\{p_i\}$, the expected total unsampled second moment after $N$ draws is

$$\sum_i p_i^2(1-p_i)^N.$$

By linearity, this ensures $U_{\text{pos},2}$ and $U_{\text{neg},2}$ decrease monotonically with $N$, driving $\Delta Q_{\text{pos}}$ toward nonnegativity as $N$ increases. A full proof is provided in Appendix A.2.

## 3. BroRL: Broad Reinforcement Learning

### 3.1. Background: Prolonged Reinforcement Learning

We adopt the prolonged reinforcement learning (RL) framework from ProRLv2 (Hu et al., 2025b). This approach is centered around a clipped Proximal Policy Optimization (PPO) algorithm, with the objective function:

$$\mathcal{L}_{\text{PPO}}(\theta) = \mathbb{E}_\tau \Big[ \min \big( r_\theta(\tau)\, A(\tau),$$
$$\text{clip}\big(r_\theta(\tau), 1 - \varepsilon_{\text{low}}, 1 + \varepsilon_{\text{high}}\big) A(\tau)\big)\Big],$$

where $r_\theta(\tau)$ is the probability ratio and $A(\tau)$ is the advantage. A key feature is its REINFORCE++-style decoupled advantage normalization (Hu et al., 2025a). First, the advantage $A_\tau$ for a trajectory $\tau$ with return $R_\tau$ is computed by subtracting the mean return of its corresponding group for each prompt. This value is then normalized across the entire global sample batch:

$$A_\tau = R_\tau - \text{mean}_{\text{group}}\big(R_\tau\big),$$
$$A_\tau^{\text{norm}} = \frac{A_\tau - \text{mean}_{\text{batch}}\big(A_\tau\big)}{\text{std}_{\text{batch}}\big(A_\tau\big)}.$$

To further improve performance and exploration, the framework integrates several key techniques. A core component is *Dynamic Sampling* (Yu et al., 2025), which filters out trivial trajectories that are either entirely correct or entirely incorrect to focus training on the most informative samples. For a batch $\mathcal{B}$ of trajectories $\tau$, the filtered batch $\mathcal{B}'$ is:

$$\mathcal{B}' = \left\{ \tau \in \mathcal{B} \;\middle|\; 0 < \sum_{i=1}^{N} \mathbb{I}(M_i = M_{\text{correct}}) < N \right\},$$

where $N$ is the number of rollout samples per prompt, $M_i$ is the prediction and $\mathbb{I}(\cdot)$ is the indicator function. Other methods include periodic resets of the reference policy, exploration enhancements via Clip-Higher ($\varepsilon_{\text{high}} > \varepsilon_{\text{low}}$) (Yu

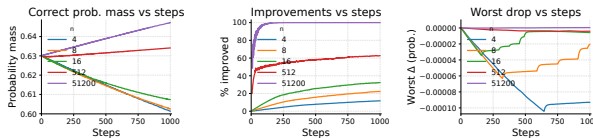

*Figure 3.* Training dynamics of the simulator under varying rollout size $N$. We track (i) the total probability mass assigned to correct actions, (ii) the fraction of correct actions whose probability increased relative to step 0, and (iii) the worst negative change in probability among correct actions. Larger $N$ produces more stable updates, faster accumulation of probability mass, and crucially it eliminates knowledge shrinkage by removing negative probability drops altogether.

et al., 2025), and truncated importance sampling (Yao et al., 2025) to correct off-policy mismatch between the inference engine and the training engine.

## 3.2. Scaling Reinforcement Learning via Number of Rollouts

BroRL is a rollout-allocation recipe rather than a new RL objective: it keeps the ProRLv2 objective, dynamic filtering, and PPO update structure fixed, but reallocates sampling compute from many prompts with few rollouts to fewer prompts with many rollouts once the small-$N$ regime has saturated. The decomposition in Theorem 1 reveals that the policy update, as measured by the change in correct probability mass ($\Delta Q_{\text{pos}}$), is subject to a potentially negative "unsampled coupling" term, $S_R\big(Q_{\text{pos}}U_{\text{neg},2} - Q_{\text{neg}}U_{\text{pos},2}\big)$, which can introduce instability and counteract policy improvement. Our theoretical framework establishes that the influence of this term shrinks as the rollout size $N$ grows under the assumptions of Lemma 2. Consequently, BroRL uses a large $N$ to increase rollout diversity for each prompt in the post-saturation regime. This rollout-size scaling can make the learning signal more reliable by reducing the variance and potential negativity arising from unsampled portions of the action space, translating our theoretical insights into a practical continuation strategy for complex reasoning tasks.

## 4. Experiments

We first conduct token-level simulations to verify our theoretical insight (Section 4.1). We then evaluate BroRL by continuing RL training from a saturated ProRLv2 checkpoint, analyzing Pass@1 training trajectories and statistical significance, breaking the steps-scaling plateau under matched compute, and studying both algorithmic and hardware efficiency; we further include ablations over rollout size $N$, extended training, score-per-token efficiency, and scaling to a 4B model (Section 4.2).

### 4.1. Simulation of the Theoretical Analysis

**Simulation Setup.** We build a token-level simulator reflecting the per-token update analysis in Theorem 1, using a TRPO-style linear surrogate objective (Schulman et al., 2015; Zhu et al., 2025). The vocabulary has size $d = 128{,}000$, with a subset $\mathcal{P} \subset [d]$ of 10,000 correct tokens assigned reward $R_i = +1$ and the remainder $R_i = -1$.

Logits $z \in \mathbb{R}^d$ are initialized as $z_i = 0$, with optional seeding by setting $z_i = 3$ for $i \in \mathcal{P}$ and fixing one anchor token $z_0 = 5$. Probabilities are $p_i = \text{softmax}(z/\tau)_i$ with $\tau = 1$. At each step $t$, we draw $N$ i.i.d. samples center rewards with the batch baseline $b = \frac{1}{N}\sum_{j=1}^{N} R_j$, yielding $\tilde{r}_j = R_j - b$ to reduce variance.

We optimize the RLVR surrogate

$$\mathcal{L}_{\text{sur}} = -\frac{1}{N}\sum_{j=1}^{N} \tilde{r}_j \, p_j$$

and update $z$ with AdamW (learning rate $\eta = 10^{-3}$) for $T = 1000$ steps. Rollout size $N \in \{4, 8, 16, 512, 51200\}$ are varied while all other hyperparameters are fixed.

After each update, we record: (i) the *total correct probability mass* $Q_{\text{pos}} = \sum_{i \in \mathcal{P}} p_i$, (ii) the *percent of correct tokens* whose probabilities increased relative to step 0, and (iii) the *worst probability drop* among correct tokens.

**Results.** The simulation results align with our key insight: increasing rollout size $N$ dampens the influence of the unsampled coupling term in Theorem 1, yielding more reliably positive mass expansion and stable policy updates. As shown in Figure 3, larger rollout size $N$ accelerates the growth of positive mass $Q_{\text{pos}}$ and increases the proportion of correct tokens whose probabilities improve at each step, whereas small-$N$ updates exhibit slower progress, higher variance, and occasional regressions.

Importantly, the worst-case probability drops among correct tokens—known as knowledge shrinkage (Wu et al., 2025) and common with small $N$— disappear at large $N$. In the extreme, when $N$ is very large, our simulator shows no knowledge shrinkage and all correct tokens gain probability mass. This matches the theoretical prediction that unsampled second-moment terms shrink with width (Lemma 2), thereby suppressing potential harmful contributions from unsampled tokens. Taken together, these findings suggest that allocating compute to rollout size, rather than step depth, can yield more consistently positive updates in this setting and provides the empirical basis for BroRL.

*Table 1.* Key BroRL training settings for LLM experiments.

| Hyperparameter | Value |
|---|---|
| Rollouts per prompt ($N$) | 512 (ablation includes 256) |
| Prompts per RL step | 128 (BroRL), 512 (ProRL baseline) |
| Learning rate | $2 \times 10^{-6}$ (scaled by $\sqrt{B}$, see text) |
| Context length | 16,384 tokens |
| KL regularization | K2/MSE KL; coefficient $2 \times 10^{-4}$ |
| Reference policy reset | about every 100 steps |

## 4.2. Empirical Study on Large Language Models

### 4.2.1. EXPERIMENTAL SETUP

**Base Model.** We build upon the publicly available Nemotron-Research-Reasoning-Qwen-1.5B model, following the ProRLv2 recipe (Hu et al., 2025b). Unless otherwise stated, our main continuation experiments start from the official ProRLv2 checkpoint after 3,000 RL steps and evaluate on five task families: math, code, science, IFEval (Zhou et al., 2023) and reasoning gym (Stojanovski et al., 2025). This 3,000-step checkpoint, trained with a context length of 8,192 tokens, provides a strong starting point. To further enhance its capabilities, especially for tasks requiring long-context reasoning, we expanded its context window to 16,384 tokens for all subsequent training phases with 64 NVIDIA H100 GPUs and the veRL framework (Sheng et al., 2025).

**BroRL Implementation.** We continue RL training on top of ProRLv2 checkpoint with the BroRL recipe. We increased the number of generated samples per prompt from the baseline of 16 to $N = 512$. This large value of $N$ is central to our hypothesis that a broader exploration of the solution space during each update step leads to more robust and generalizable reasoning abilities. For baseline comparison, we also continue RL training from the same checkpoint using the original ProRL recipe ($N$=16), and compare checkpoints at comparable wall-clock compute (Table 2).

**Training Data.** We follow ProRLv2's multi-domain RLVR training mixture (about 136k prompts) spanning math, code, STEM, logical puzzles, and instruction-following. Concretely, we use: DeepScaleR (math) (agentica-org, 2025); Eurus-2 RL (code) (PRIME-RL, 2025); SCP-116K (STEM) (EricLu, 2025); Reasoning Gym (puzzles) (open-thought, 2025); and the Llama-Nemotron post-training RL split (instruction-following) (NVIDIA, 2025).

**Key Training Hyperparameters.** Table 1 lists the key settings we used for BroRL continuation.

**Learning Rate Scaling.** To maintain training stability while accommodating the significantly larger effective batch size resulting from the increased rollout size ($N$), we adjusted the learning rate while keeping the number of PPO mini-batches per step unchanged. Specifically, the learning rate was scaled proportionally to the square root of the increase in the batch size (Krizhevsky, 2014). Let $\eta_0$ be the base learning rate for a reference batch size $B_0$. Our new learning rate $\eta_{\text{new}}$ for a new, larger batch size $B_{\text{new}}$ is determined by the formula:

$$\eta_{\text{new}} = \eta_0 \times \sqrt{\frac{B_{\text{new}}}{B_0}}.$$

This principled adjustment ensures that the magnitude of parameter updates remains well-controlled.

### 4.2.2. ANALYSIS OF PASS@1 SUCCESS RATE

Representative trajectories are provided in Appendix Figure 4.

To better understand the practical impact of our methods, we compare BroRL and ProRL across benchmark tasks under equalized training compute. Figure 4 summarizes these results by tracking performance at intermediate checkpoints. We observe three characteristic types of training trajectories. In the first, both methods improve but BroRL consistently outperforms ProRL, aligning with theoretical expectations and highlighting stronger learning dynamics. In the second, ProRL degrades over time while BroRL continues to improve, underscoring its robustness. In the third, both methods fail to achieve consistent gains, suggesting that $N = 512$ might not be large enough for some of the harder problems. Most benchmarks fall into the first two patterns, while the third is less common. Collectively, these trajectories show that BroRL not only matches theoretical predictions but also demonstrates clear practical advantages in training efficiency and stability. Importantly, all results are measured on the test dataset, highlighting that BroRL's improvements reflect not only better learning dynamics during training but also stronger generalization to unseen instances.

To complement the trajectory analysis, we perform a statistical evaluation to test whether BroRL provides a measurable improvement over ProRL. We collect results from all individual problem instances across benchmarks, yielding over 10,000 data points, and measure Pass@1 at the final checkpoint under equal training compute ($\sim 140$ hours). A paired t-test reveals a small but statistically significant advantage for BroRL ($\Delta = 0.0033$, $t = 4.84$, one-tailed $p = 6.5 \times 10^{-7}$). A one-tailed paired t-test rejects the null hypothesis, confirming that BroRL outperforms ProRL with extremely strong statistical confidence. Although the mean difference is small, this is expected since we build on a strong baseline already fine-tuned for 3000 steps and evaluate after only 100 additional steps, where gains scale roughly log-linearly with training time (Liu et al., 2025a).

In this regime, even a modest but statistically significant improvement is meaningful, confirming that BroRL yields more reliable progress and better generalization to unseen test instances.

### 4.2.3. PUSHING REASONING BOUNDARIES BEYOND STEPS SCALING

A common challenge in long-term RLVR training is performance saturation, where simply training longer steps yields diminishing returns. The initial ProRLv2 checkpoint trained 3000 RL steps had reached such a plateau. This section details a controlled experiment to demonstrate that BroRL's rollout-scaling approach is not only more effective but also more time-efficient at breaking through this performance ceiling.

We compare two continuation strategies from the same saturated 3k-step ProRLv2 checkpoint. The ProRL baseline continues training with a conventional small rollout size ($N = 16$) and 512 prompts per RL step. Our BroRL continuation instead scales the rollout size to $N = 512$ while reducing prompts per step to 128, and we report checkpoints that are comparable in wall-clock compute. Table 2 and Figure 1 summarize the trade-offs in computational cost and benchmark outcomes. For completeness, we also include one longer BroRL continuation checkpoint (+419) that uses substantially more wall-clock time, to illustrate continued gains when training longer with large-$N$ rollouts.

The result reveals two divergent outcomes. The ProRL method shows marginal initial gains across all tasks, peaking at 62.08 on Math and 62.10 on Reasoning Gym. However, continued training leads to performance stagnation and degradation. While the Code Score sees a minor increase to 52.74, the Math Score drops to 62.02, and the Reasoning Gym Score falls significantly to 61.45. This pattern, observed after nearly 134 hours, clearly illustrates the diminishing and ultimately negative returns of simply scaling training steps for this saturated model.

In stark contrast, the BroRL approach demonstrates robust and continuous improvement across all three benchmarks, ultimately achieving the highest scores: 63.03 in Math, 54.20 in Code, and 63.09 in Reasoning Gym. The efficiency of this rollout-size-scaling approach is particularly striking. After just 98.1 hours, BroRL had already decisively surpassed the final performance of the ProRL method across all metrics, doing so in approximately 35 fewer hours. This result confirms that scaling the rollout size $N$ is a more effective and computationally efficient strategy for pushing the performance boundaries of a saturated model. This superior performance stems not from performing more gradient updates, but from executing fewer, yet higher-quality updates, as we maintain the same number of PPO minibatches per RL step. More evaluation details and results are

in Appendix B. The following section investigates the core reasons for this enhanced efficiency at both the algorithmic and hardware levels.

### 4.2.4. SCALING TO A LARGER (4B) MODEL

To test whether the training stability benefit of BroRL persists at larger scale, we ran additional experiments on a 4B instruction-tuned model. Table 3 shows that continued ProRL training after saturation can lead to a noticeable math drop, while BroRL avoids significant regressions and maintains a roughly monotonic trajectory.

### 4.2.5. IMPACT OF ROLLOUT SIZE SCALING ON GPU COMPUTE EFFICIENCY

The primary performance bottleneck in training models for long Chain-of-Thought (CoT) reasoning via RLVR is the sample generation phase (Hu et al., 2024). Our BroRL framework addresses this challenge through a two-pronged approach: one at the algorithmic level and another at the hardware level. To isolate these variables, all experiments were conducted on an identical hardware setup (GPU and node count). Table 4 quantifies these two factors.

First, at the algorithmic level, a larger rollout size $N$ leads to a more diverse set of candidate samples. The *Dynamic Sampling Pass Rate* in Table 4 shows that with $N = 512$, 62% of the generated samples are deemed useful for training, compared to only 41% for $N = 16$. This minimizes wasted computation and ensures each training step is based on more effective data.

Second, at the hardware level, our approach achieves a significantly higher generation throughput—nearly 100% faster (72.4 vs 36.5 samples/s) in our setup. This improvement is consistent with alleviating a common GPU inference bottleneck: memory-bound execution (Recasens et al., 2025). With small batches ($N = 16$), generation is often memory-bound; the GPU's compute cores can idle while waiting to fetch data from memory. By generating a large number of samples ($N = 512$) at once, the operation can move toward a more compute-bound regime and may also benefit from a higher prefix cache hit rate (Zheng et al., 2024). Overall, BroRL not only improves training outcomes in our experiments but also utilizes the underlying hardware more efficiently in this deployment setting.

### 4.2.6. ABLATION ON ROLLOUT SIZE $N$

To address the question of how performance scales with intermediate rollout sizes, we include an additional ablation point at $N = 256$. Table 5 shows a consistent positive trend across all three benchmarks: $N = 512 > N = 256 > N = 16$ under comparable continuation budgets.

*Table 2.* Efficiency and Performance Comparison. BroRL shows steady improvement and achieves a higher score in less total time, while the ProRL stagnates and degrades. The number of samples refers to the amount before dynamic sampling filtering.

| Method | N | Prompts / Step | RL Steps | Samples (k) | Time (h) | Math Score | Code Score | Reasoning Gym Score |
|---|---|---|---|---|---|---|---|---|
| Baseline | 16 | 512 | 3000 | - | - | 61.69 | 52.00 | 61.29 |
| ProRL | 16 | 512 | +225 | +4390 | 56.3 | 62.08 | 52.26 | 62.10 |
| ProRL | 16 | 512 | +535 | +10439 | 133.8 | 62.02 | 52.74 | 61.45 |
| BroRL | 512 | 128 | +107 | +11226 | 98.1 | 62.62 | 53.31 | 62.71 |
| BroRL | 512 | 128 | +134 | +14059 | 122.8 | 62.85 | 53.48 | 62.82 |
| BroRL | 512 | 128 | +191 | +20039 | 173.8 | 63.03 | 54.20 | 63.09 |
| BroRL | 512 | 128 | +419 | +45416 | 393.9 | **63.66** | **56.64** | **63.40** |

*Table 3.* Results on a 4B model (Qwen3-4B-Instruct). BroRL improves stability after saturation.

| Method | RL Steps | Math | Code | RG |
|---|---|---|---|---|
| Qwen3-4B-Instruct-2507 | 0 | 68.22 | 51.77 | 28.97 |
| ProRLv2 ($N$=16, $B$=512) | 333 | 71.65 | 64.79 | 77.95 |
| ProRLv2 ($N$=16, $B$=512) | 396 | 69.58 | 66.15 | 78.15 |
| BroRL ($N$=256, $B$=128) | 333+30 | 71.64 | 65.54 | 78.13 |

*Table 4.* Algorithmic and hardware efficiency metrics.

| Method (N) | Dyn. samp. pass rate | Gen. throughput (samples/s) |
|---|---|---|
| ProRL (16) | 41% | 36.5 |
| BroRL (512) | 62% | 72.4 |

*Table 5.* Ablation on rollout size $N$ (continuing from the 3k-step checkpoint). We report average scores; other settings follow ProRLv2.

| Method | N | RL Steps | Time (h) | Math / Code / RG |
|---|---|---|---|---|
| ProRL | 16 | 535 | 133.8 | 62.02 / 52.74 / 61.45 |
| BroRL | 256 | 202 | 118.2 | 62.23 / 53.43 / 61.78 |
| BroRL | 512 | 134 | 122.8 | 62.85 / 53.48 / 62.82 |

*Table 6.* Score-per-token efficiency (ProRL vs. BroRL).

| Task | ProRL | BroRL | Diff | ProRL tokens | BroRL tokens | Token diff |
|---|---|---|---|---|---|---|
| Math | 62.02 | 63.66 | +1.64 | 16,506 | 15,760 | -745 |
| Code | 52.74 | 56.64 | +3.90 | 26,808 | 26,090 | -717 |

### 4.2.7. SCORE-PER-TOKEN EFFICIENCY

We also find that BroRL attains higher scores with fewer generated tokens on both math and code benchmarks. Table 6 reports the aggregate score and the average total output tokens under the same evaluation protocol.

## 5. Discussion and Limitations

**When to use large $N$.** Our experiments support a narrower claim than "larger $N$ is always better": large-$N$ continuation is effective after ProRL-style small-$N$ training has already saturated. A practical heuristic is to train with smaller $N$ while validation performance improves, then increase $N$ when progress flattens and the generation system can benefit from larger batch width. Under a fixed compute budget, the optimal allocation trades off rollout size, prompts per step, and number of update steps. Our $N = 256$ ablation and the concurrent IsoCompute analysis (Cheng et al., 2026) both suggest diminishing returns rather than unbounded gains from increasing $N$.

**Adaptive rollout allocation.** The positivity margin implicit in Theorem 1 could be used as a difficulty-aware signal for assigning more rollouts to prompts with smaller or negative expected progress. We do not validate such

an adaptive-$N$ algorithm in this paper. Methods such as DARS and AR3PO already study difficulty-adaptive allocation and response reuse (Yang et al., 2025; Zhang et al., 2025); our analysis is complementary because it explains why broader observed support can reduce finite-sample unsampled coupling, while those methods study where rollout budget should be spent most efficiently.

**Scope.** Our empirical evidence covers math, code, and multi-domain reasoning benchmarks, but not interactive multi-turn environments with intermediate feedback. Large rollout groups also require substantial generation memory and scheduling capacity; the nearly doubled throughput we observe is specific to our hardware and inference setup. Practitioners with smaller GPU pools may need to use intermediate $N$, generation accumulation, or adaptive allocation rather than $N = 512$ uniformly.

## 6. Related Work

**Reinforcement Learning for Reasoning** Reasoning models often generate long chains of thought before producing an answer. This training paradigm is central to systems such as DeepSeek-R1 (Guo et al., 2025), and RLVR adapts RLHF techniques (Christiano et al., 2017; Ouyang et al., 2022) by replacing subjective human feedback with verifiable correctness rewards. This line has popularized GRPO (Shao et al.,

2024), RLOO (Ahmadian et al., 2024), REINFORCE++ (Hu et al., 2025a), DAPO (Yu et al., 2025), and prolonged-training recipes such as ProRL (Liu et al., 2025a). Concurrent systems such as JustRL (He et al., 2025) and QuestA (Li et al., 2025) emphasize simpler stable recipes and question augmentation. BroRL targets a different regime: continuing a strong ProRL checkpoint after small-$N$ step scaling has saturated.

**Rollout Count and Sampling Error**    Several recent analyses study how rollout count, batch size, and baselines affect RL fine-tuning. Xie et al. (2026) analyze design choices from a batched contextual-bandit perspective, while Zeng et al. (2025) study shrinkage baselines that reduce policy-gradient variance in low-generation regimes. Cheng et al. (2026) explicitly optimize the allocation among parallel roll-outs, problem batch size, and update steps, finding that the compute-optimal number of rollouts grows with budget and then saturates. In broader RL, Corrado & Hanna (2026) describe finite-sample "sampling error," where sampled trajectories deviate from the expected on-policy distribution. Our "unsampled coupling" term can be viewed as a specific RLVR token-level manifestation of this finite-sample mismatch: it isolates how probability mass outside the sampled rollout set can oppose correct-mass expansion.

**Adaptive and Diversity-Oriented Rollouts**    Broader sampling is complementary to methods that decide where roll-out budget should be allocated or which samples should be used. DARS (Yang et al., 2025) and AR3PO (Zhang et al., 2025) allocate more rollouts to difficult prompts; EFRame (Wang et al., 2025) adds exploration-filter-replay stages; and PODS (Xu et al., 2025) generates many rollouts but updates on a diverse subset. Other work emphasizes entropy and rollout diversity as stabilizers, including M-GRPO (Bai et al., 2025) and the entropy mechanism of RL for reasoning (Cui et al., 2025). These observations also explain why uniformly increasing $N$ need not help in every regime. BroRL contributes a complementary mechanism-level account and empirical evidence for the post-saturation setting, while adaptive allocation and diversity-preserving selection remain natural extensions.

## 7. Conclusion

This work establishes rollout size $N$, not just longer steps, as a critical and efficient axis for scaling reinforcement learning in large language models. We showed that the performance plateaus encountered by steps-scaling methods like ProRL can reflect limitations of the learning signal under insufficient exploration. Our theoretical analysis pinpointed the "unsampled coupling" term as a source of this instability and showed that increasing rollout size $N$ mitigates its impact under our assumptions. Empirically, our BroRL framework

improved a saturated model and yielded continued gains on complex reasoning tasks for a 1.5B model in our evaluation. Critically, these gains were achieved with improved computational efficiency; in our hardware setup we observed nearly doubled generation throughput, consistent with shifting the bottleneck from memory toward compute in some cases.

## Impact Statement

This work introduces rollout-size scaling as a principled and efficient axis for reinforcement learning, enabling continued improvements in reasoning models beyond step-scaling plateaus. By improving sample efficiency and hardware utilization, our approach can lower the computational barrier to training strong reasoning systems, benefiting research and education. At the same time, stronger reasoning and code-generation capabilities may amplify downstream risks if misused, underscoring the importance of responsible deployment and continued investment in AI safety and governance.

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

# A. Proof Details

## A.1. Theorem 1

**Notation.** For clarity, we repeat key quantities: (i) $A, B, U$: sampled correct, sampled incorrect, and unsampled token sets. (ii) $Q_{\text{pos}}, Q_{\text{neg}}$: global correct/incorrect probability masses. (iii) $A_2, B_2, U_{\text{pos},2}, U_{\text{neg},2}$: second moments. (iv) . $S_R = R_c\, P_{\text{pos}} + R_w\, P_{\text{neg}}$ which represents the net contribution of sampled tokens, balancing the rewards from correct and incorrect tokens. Define the reward $R_j$ for sampled correct, sampled incorrect and unsampled tokens as:

$$
R_j = \begin{cases} R_c, & j \in A, \\ R_w, & j \in B, \\ 0, & j \in U. \end{cases}
$$

**Logit update and Jacobian expansion.** We start from the TRPO-style (Schulman et al., 2015) linear surrogate

$$
L_{\text{RLVR}}(\theta) = -\mathbb{E}_{x \sim \mathcal{D}} \Big[ \sum_y r(x,y)\, \pi_\theta(y \mid x) \Big]
$$

$$
\approx -\frac{1}{N} \sum_{i \in A \cup B \cup U} R_i p_i,
$$

where $R_i \in \{R_w, 0, R_c\}$. This linear surrogate furnishes a convenient Monte-Carlo estimate - sample average approximation when using a relative entropy - Kullback-Leibler regularizer. This estimator is unbiased, hence all derivation and integration operations carry through to be interchanged with the expectation sign (Asmussen & Glynn, 2007).

Denote $z_j$ as the logit for the $j$-th token. Then we differentiating w.r.t. $z_j$ using $\frac{\partial p_i}{\partial z_j} = p_i(\delta_{ij} - p_j)$ gives

$$
\Delta z_j = \frac{\eta}{N} p_j (R_j - S_R), \quad S_R = R_c\, P_{\text{pos}} + R_w\, P_{\text{neg}}
$$

**First-order change in probabilities.** By first-order expansion,

$$
\Delta p_i := \sum_{j=1}^{V} \frac{\partial p_i}{\partial z_j} \Delta z_j := p_i \Big( \Delta z_i - \sum_{j=1}^{V} p_j \Delta z_j \Big).
$$

Summing over any index set $\mathcal{S}$,

$$
\sum_{i \in \mathcal{S}} \Delta p_i := \sum_{i \in \mathcal{S}} p_i \Delta z_i : - \Big( \sum_{i \in \mathcal{S}} p_i \Big) \Big( \sum_{j=1}^{V} p_j \Delta z_j \Big).
$$

We will need

$$
\sum_{j=1}^{V} p_j \Delta z_j := \frac{\eta}{N} \Big[ (R_c - S_R) A_2 + (R_w - S_R) B_2 - S_R\, U_2 \Big],
$$

and, restricted to correct tokens,

$$
\sum_{i \in \mathcal{P}} p_i \Delta z_i := \frac{\eta}{N} \Big[ (R_c - S_R) A_2 - S_R\, U_{\text{pos},2} \Big].
$$

**Total change of correct mass.** The total change of correct-token probability mass is

$$
\Delta P_{\text{correct}} \equiv \sum_{i \in \mathcal{P}} \Delta p_i := \sum_{i \in \mathcal{P}} p_i \Delta z_i : -Q_{\text{pos}} \sum_{j=1}^{V} p_j \Delta z_j.
$$

Substituting the identities above and simplifying with

$$
\begin{aligned}
Q_{\text{pos}} &= P_{\text{pos}} + P_{\text{pos,out}}, \\
Q_{\text{neg}} &= 1 - Q_{\text{pos}}, \\
U_2 &= U_{\text{pos},2} + U_{\text{neg},2},
\end{aligned}
$$

we obtain the compact form

$$
\begin{aligned}
\Delta P_{\text{correct}} := \frac{\eta}{N} \Big[ &(R_c - S_R)\, Q_{\text{neg}}\, A_2 \\
&+ (S_R - R_w)\, Q_{\text{pos}}\, B_2 \\
&+ S_R \Big( Q_{\text{pos}}\, U_{\text{neg},2} - Q_{\text{neg}}\, U_{\text{pos},2} \Big) \Big],
\end{aligned}
\tag{2}
$$

with $S_R = R_c P_{\text{pos}} + R_w P_{\text{neg}}$.

## A.2. Lemma 2

We seek to obtain the scaling of the the unsampled second-moment with respect to $N$. For this, we work under the simple assumption of token drawn independently and identically distributed as Bernoulli random variables. This is a popular assumption (see e.g. Du et al. (2025)), which allows us to obtain a convenient analytical formula capturing the scaling we are interested in. This scaling is further corroborated by the extensive experimental results from Section 4.

Let $X \sim \text{Bin}(N, p)$ be the number of times a token is drawn in $N$ independent Bernoulli trials, each with success probability $p$. By the binomial distribution, the probability of never drawing the token is

$$
\Pr[X = 0] = (1 - p)^N.
$$

Equivalently, by independence across draws, the probability that the token is not selected in any of the $N$ trials is also $(1 - p)^N$. Define the indicator variable $I = \mathbf{1}\{X = 0\}$, which is 1 if the token is never sampled and 0 otherwise. The token's unsampled second-moment contribution is then the random variable

$$
S = p^2 I.
$$

Taking expectations, we obtain

$$
\begin{aligned}
\mathbb{E}[S] &= p^2\, \mathbb{E}[I] \\
&= p^2\, \Pr[I = 1] \\
&= p^2\, \Pr[X = 0] \\
&= p^2 (1 - p)^N.
\end{aligned}
$$

# B. Empirical Evaluation

To rigorously test whether rollout size $N$ scaling breaks the training–depth plateau observed at 3,000 RL steps in the baseline (Liu et al., 2025a), we compare *ProRL* (small rollout size $N = 16$ and longer steps) against *BroRL* (large rollout size $N = 512$) under an identical evaluation protocol across three task families: math competitions (AIME/AMC, MATH, Minerva, OlympiadBench (Mathematical Association of America, 2026a;b; Hendrycks et al., 2021b; Lewkowycz et al., 2022; He et al., 2024)), code generation (APPS, CodeContests/Codeforces, TACO (Hendrycks et al., 2021a; Li et al., 2022; 2023)), and multi-domain reasoning (Reasoning Gym (Stojanovski et al., 2025)). Importantly, the table columns capture *training* controls: $N$ is the number of samples per prompt, $B$ is the number of prompts per RL step, and Steps is the count of continued RL steps. For details on sample generation and GPU compute consumption, please refer to Table 2. For evaluation, we report **pass@1** with a 32k context length, averaged over 16 independent samples per instance to ensure stable estimates, using nucleus sampling (top_p=0.95) with a temperature of 0.6.

The experimental results presented in Tables 7, 8, and 9 consistently support the benefit of rollout size $N$ scaling in our setting. In the critical domain of mathematical reasoning, the *ProRL* approach exhibits a plateau; after an initial small gain (from a 61.69 baseline to 62.08), its average score slightly degrades to 62.02. In contrast, *BroRL* avoids this degradation and improves, reaching 62.85 after 134 steps. This advantage is also observed in other domains. For code generation, BroRL's score increase (+1.48 points) exceeds the gains from ProRL (+0.74 points). Similarly, on the Reasoning Gym benchmark, BroRL improves by over 1.5 points, while ProRL provides little gain under the same protocol.

In conclusion, across all three demanding domains, widening the generation search space per RL step proves to be a significantly more effective and efficient strategy than merely continuing training with a narrow search. Crucially, Table 2 shows that at comparable wall-clock time, BroRL achieves higher scores across Math/Code/Reasoning Gym (e.g., BroRL +134 at 122.8 hours surpasses ProRL +535 at 133.8 hours), highlighting that the diversity of experience in each training step can be more important than simply training longer with small-$N$ rollouts.

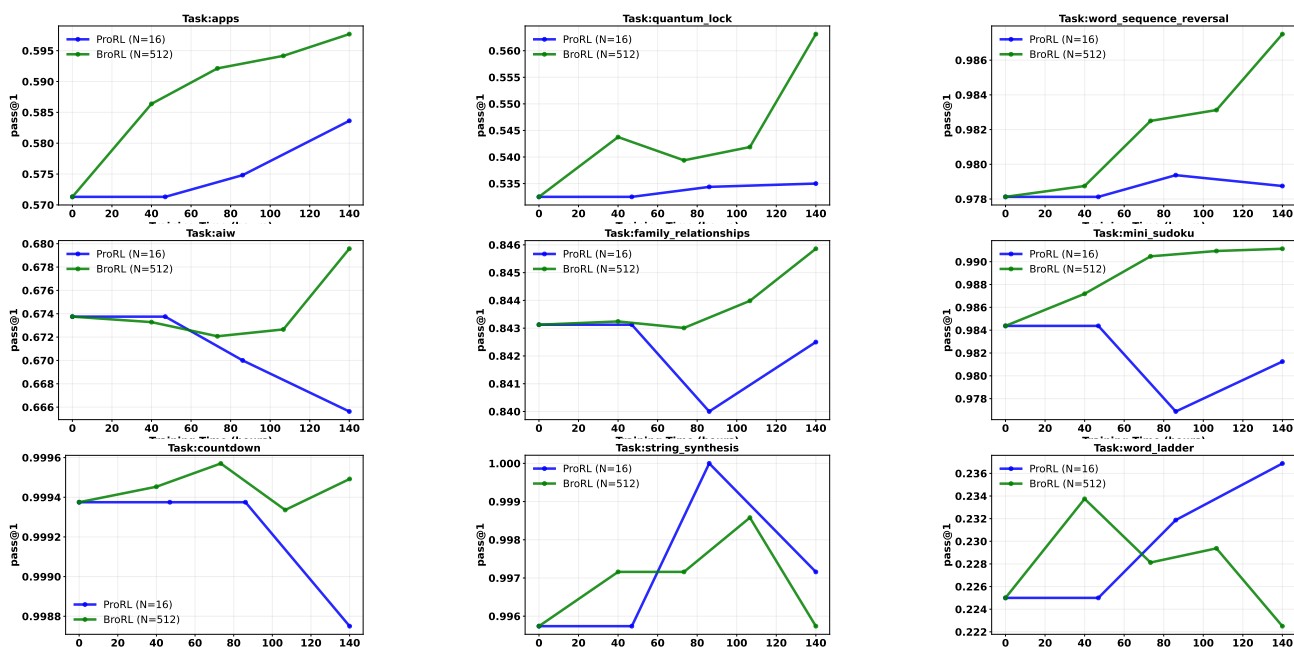

*Figure 4.* Pass@1 comparison of BroRL vs. ProRL, normalized by training compute. Rows show representative trajectories: (1) both improve but BroRL consistently outperforms ProRL; (2) ProRL degrades while BroRL continues to improve; (3) both methods fail to yield consistent gains.

*Table 7.* Math scores.

| Method | N | B | Steps | AIME24 | AIME25 | AMC | Math | Minerva | Olympiad Bench | Math Avg. |
|---|---|---|---|---|---|---|---|---|---|---|
| Baseline | 16 | 512 | 3000 | 49.58 | 36.04 | **82.53** | **92.49** | 49.03 | 60.44 | 61.69 |
| ProRL | 16 | 512 | +225 | 54.58 | 36.25 | 80.95 | 91.93 | 48.25 | 60.52 | 62.08 |
| ProRL | 16 | 512 | +535 | 54.38 | 35.83 | 80.42 | 92.15 | 48.55 | 60.77 | 62.02 |
| BroRL | 512 | 128 | +107 | 56.10 | 35.30 | 81.76 | 92.18 | 48.92 | 61.41 | 62.62 |
| BroRL | 512 | 128 | +134 | **57.71** | 35.63 | 80.12 | 92.06 | **49.72** | **61.87** | 62.85 |
| BroRL | 512 | 128 | +191 | 57.50 | **36.88** | 81.02 | 92.14 | 49.08 | 61.54 | **63.03** |

*Table 8.* Code generation scores.

| Method | N | B | Steps | apps | codecontests | codeforces | taco | Code Avg. |
|---|---|---|---|---|---|---|---|---|
| Baseline | 16 | 512 | 3000 | 58.52 | 54.99 | 58.64 | 35.87 | 52.00 |
| ProRL | 16 | 512 | +225 | 58.83 | 54.58 | 59.27 | 36.36 | 52.26 |
| ProRL | 16 | 512 | +535 | 59.67 | 55.09 | 59.13 | 37.06 | 52.74 |
| BroRL | 512 | 128 | +107 | 60.28 | 55.84 | 59.80 | 37.31 | 53.31 |
| BroRL | 512 | 128 | +134 | 60.19 | 56.52 | 60.04 | 37.15 | 53.48 |
| BroRL | 512 | 128 | +191 | **61.59** | **56.62** | **60.86** | **37.74** | **54.20** |

*Table 9.* Reasoning Gym scores.

| Method | N | B | Steps | algebra | algorithmic | arc | arithmetic | code | cognition | games | geometry | graphs | induction | logic | Avg. |
|---|---|---|---|---|---|---|---|---|---|---|---|---|---|---|---|
| Baseline | 16 | 512 | 3000 | 97.19 | 55.32 | 4.98 | 85.74 | 48.20 | 45.91 | 25.68 | 91.62 | 70.25 | 80.25 | 82.25 | 61.29 |
| ProRL | 16 | 512 | +225 | 97.01 | 58.22 | 5.33 | 85.74 | 47.96 | 46.01 | 25.55 | 91.59 | 69.83 | 80.31 | 85.26 | 62.10 |
| ProRL | 16 | 512 | +535 | 97.46 | 55.56 | 4.79 | 85.70 | 48.43 | **46.33** | 25.71 | 92.56 | 70.40 | 80.31 | 85.29 | 61.45 |
| BroRL | 512 | 128 | +107 | 97.55 | 59.11 | 5.10 | 85.97 | 49.22 | 44.05 | **25.99** | 92.16 | 71.51 | 80.40 | 85.41 | 62.71 |
| BroRL | 512 | 128 | +134 | **97.70** | 59.28 | 5.31 | 85.95 | 49.30 | 44.53 | 25.88 | 92.88 | 72.01 | 80.38 | 85.29 | 62.82 |
| BroRL | 512 | 128 | +191 | 97.59 | **59.65** | **6.27** | **86.17** | **49.45** | 45.51 | 25.77 | **93.00** | **72.03** | **80.94** | **85.56** | **63.09** |

