# OpenReview forum: "BroRL: Scaling Reinforcement Learning via Broadened Exploration"
_ICML.cc/2026/Conference — ICML 2026 regular_

### Official Review · Reviewer_98hh · 2026-02-26

**Soundness:** 3
**Presentation:** 3
**Significance:** 3
**Originality:** 3
**Overall Recommendation:** 4
**Confidence:** 3

**Summary:**

This paper proposes BroRL to address the performance plateau and "knowledge shrinkage" in RLVR by scaling up the number of rollout samples per prompt $N$. By exhaustively broadening exploration through larger $N$, BroRL theoretically diminishes the negative coupling effects of unsampled tokens, ensuring a strictly positive accumulation of probability mass for correct responses. Furthermore, this large-$N$ approach can potentially improve computational efficiency by shifting the generation process from memory-bound to compute-bound. Experiments demonstrate that BroRL successfully revives 1.5B models stagnated after 3K training steps, achieving continuous improvements across benchmarks.

**Compliance With Llm Reviewing Policy:**

Affirmed.

**Final Justification:**

Overall, the authors’ explanation addresses my concern to some extent, and I am hesitating between a score of 3 and 4.

I still believe Q2 urgently needs controlled comparisons and empirical results. Since Q2 is linked to one of the paper’s main contributions, it is hard to justify the lack of more extensive comparisons and ablation studies. I strongly encourage the authors to include such analyses in the next revision to make the paper more rigorous and convincing.

Compared with a 3, I am ultimately leaning toward a 4. However, I do think this paper falls somewhere between a 3 and 4: the conclusions are somewhat too simple, there is insufficient further extension and discussion, and some of the proposed extensions lack corresponding empirical support and comparison.

**Key Questions For Authors:**

Q1: At what point do the performance gains from increasing $N$ saturate? Given a fixed compute budget, does a larger $N$ strictly guarantee better final performance, or is there an optimal trade-off between rollout size $N$, prompts per step (batch size), and total RL training steps?

Q2: Regarding the transition from a smaller $N$ to a larger $N$, what is the optimal timing or criteria for making this switch during training? Could the authors provide further analysis or heuristic guidelines on this?

Q3: Is it feasible to dynamically allocate the number of rollouts $N$ based on the current prompt's difficulty? And a deeper discussion on how problem complexity (easy vs. hard) influences the optimal choice of $N$ would be great.

Q4: If evaluated on non-math domains or multi-turn tasks, would the scaling behavior and effectiveness of $N$ remain consistent with the current findings?

I am willing to raise my score if these questions(especially Q1, Q2, and Q3) are adequately addressed.

**Limitations:**

yes

**Strengths And Weaknesses:**

### Strengths:

S1: Scaling up the group size (i.e., rollout samples per prompt, $N$) is a simple yet highly effective approach.

S2: The theoretical analysis regarding the change in correct-token probability mass provides a reasonable explanation for why a larger $N$ yields better performance.

S3: The experimental results are convincing and adequately demonstrate the effectiveness of scaling $N$ on MATH reasoning tasks.

### Weaknesses:

W1: The main conclusion that simply increasing $N$ improves performance is a bit trivial, as the benefits of larger rollout sizes are already somewhat well recognized in existing literature. The paper would be significantly stronger if it provided actionable insights on how to determine the optimal $N$ under a fixed compute budget. Intuitively, adjusting $N$ involves critical trade-offs with batch size(Prompts/Step) and total training steps(e.g., see [Ref]).

W2: The experiments primarily focus on single-turn tasks (mostly math-oriented). The evaluation would be more comprehensive if it included diverse domains, particularly multi-turn tasks. Multi-turn environments involve intermediate feedback and inherently have higher trajectory diversity, which might exhibit different scaling properties regarding $N$.

[Ref] IsoCompute Playbook: Optimally Scaling Sampling Compute for RL Training of LLMs.

---

> ### Author Rebuttal · Authors · 2026-03-29
>
> We thank the reviewer for the careful reading and constructive questions. We are encouraged that the reviewer finds the method effective, the theory reasonable, and the experiments convincing. We address Q1-Q3 in detail below.
>
> ##  Q1: Trade-off among N, batch size, and training steps under fixed compute
>
> We agree this is the central practical question. The concurrent IsoCompute Playbook (Cheng et al., 2026) studies exactly this: compute-optimal allocation of sampling compute for on-policy RL in LLMs, across ~120,000 H200-hours. Their findings are consistent with ours:
>
> The compute-optimal number of parallel rollouts increases predictably with compute budget, then saturates.
> Different mechanisms drive the benefit on easy vs hard problems: solution sharpening on easy problems, coverage expansion on hard ones.
> Increasing rollouts mitigates interference across problems, while batch size mainly affects stability within a broad range.
>
> Our simulations (Figure 3) show a similar concave shape, suggesting diminishing returns at very large N. Our contribution relative to IsoCompute is the theoretical mechanism explaining why larger N helps (the unsampled coupling term in Theorem 1). In the revision, we will cite IsoCompute, clarify that gains from increasing N are not unbounded, and position BroRL as targeting the regime where standard RLVR has already saturated.
>
> ##  Q2: When to switch from smaller N to larger N
>
> Our current criterion is engineering-driven: increase N when the current setup no longer saturates GPU decoding throughput. The heuristic is: start with smaller N while training progresses well; monitor validation improvement and hardware utilization; when gains flatten and there is room to better saturate compute, increase N.
>
> The theoretical margin M(N) from our analysis (described in Q3 below) could serve as a more principled signal: when M(N) is comfortably positive for most prompts, the current N is sufficient; when M(N) becomes small or negative for a substantial fraction of prompts, increasing N would help. We will add both guidelines to the revision.
>
> ##  Q3: Dynamic allocation of N based on prompt difficulty
>
> This is a promising direction, we can implement a dynamic group size allocation mechanism where the rollout count G is adjusted per-prompt based on the current solve rate observed during training. Specifically, we compute a difficulty signal from the fraction of correct rollouts for each prompt, and allocate more rollouts to prompts where the solve rate falls in a range that benefits most from additional exploration (neither trivially easy nor impossibly hard).
>
> Our theory already provides the mechanism for this. The positivity margin from Theorem 1,
>
> M(N) = (R_c - S_R) Q_neg A_2 + (S_R - R_w) Q_pos B_2 + S_R(Q_pos U_neg,2 - Q_neg U_pos,2)
>
> directly measures whether the RLVR update will increase correct-token probability mass for a given prompt at a given N. Hard prompts (low accuracy) have smaller A_2 (fewer sampled-correct tokens), larger U_pos,2 (more correct tokens remain unsampled), and often negative batch reward, producing smaller or negative M(N). Easy prompts (high accuracy) have large A_2 and small unsampled terms, yielding large positive M(N). So an adaptive rule based on M(N) automatically assigns more rollouts to hard prompts and fewer to easy ones, achieving difficulty-based dynamic allocation grounded in theory.
>
> This connects directly to DARS (Yang et al.) and AR3PO (Zhang et al.), which implement difficulty-adaptive allocation using empirical accuracy as the difficulty signal. Under a fixed compute budget, DARS shows that concentrating rollouts on harder prompts is more efficient than uniform allocation. Our M(N) provides a theoretically motivated alternative that accounts not just for accuracy but also for the concentration of sampled and unsampled probability mass. We will present the dynamic allocation mechanism with preliminary results and discuss its relationship to these adaptive methods.
>
> ##  Q4: Non-math and multi-turn tasks
>
> BroRL already evaluates on code generation (APPS, CodeContests, Codeforces, TACO) and reasoning gym (11 subcategories including algebra, algorithmic, arc, cognition, games, geometry, graphs, induction, logic). The consistent improvement across these varied tasks suggests the benefit is not math-specific. For multi-turn tasks, we expect the qualitative trend to hold since our models already produce long reasoning chains with substantial sequence-level diversity. We will note multi-turn evaluation as an explicit next step.

---

> > ### Author Rebuttal · Reviewer_98hh · 2026-04-02
> >
> > Thank you for the clarifications. I would appreciate it if the authors could provide more empirical evidence or some direct comparisons regarding Q2 and Q3 to substantiate the rebuttal's arguments.

---

> > > ### Author Response · Authors · 2026-04-02
> > >
> > > - Thank you for the follow-up. We agree with the reviewer that Q2 and Q3 would ideally be supported by direct controlled comparisons. In the current submission, however, our empirical evidence is strongest for a narrower claim: **large-(N) continuation is particularly effective once standard step-scaling has already plateaued**. Specifically, starting from the saturated 3k-step ProRL checkpoint, increasing rollout width yields better performance and wall-clock efficiency than continuing with small (N). This is the regime directly supported by our current experiments.
> > >
> > > - For **Q2**, our choice to increase (N) in the later stage was motivated by practical training observations and compute considerations, rather than by a dedicated timing ablation included in the paper. In particular, our working hypothesis is that applying a very large (N) from the beginning is not necessarily the most compute-efficient strategy, whereas increasing (N) becomes more useful after training has entered a saturation regime. Since we do not currently include a controlled early-vs-late switching comparison in the submission, we will revise the paper to present this as a practical heuristic/design choice rather than a proven optimal scheduling rule.
> > >
> > > - For **Q3**, our intuition is similarly that, later in training, many easier prompts are already close to saturation while harder prompts remain exploration-limited, so additional rollouts are more likely to help the latter. However, we agree that this is not yet directly established by the current experiments as a difficulty-adaptive allocation result. We will therefore revise the wording to frame this as a plausible explanation and promising future direction, rather than a validated empirical conclusion.

---

### Official Review · Reviewer_BUL1 · 2026-03-13

**Soundness:** 3
**Presentation:** 3
**Significance:** 2
**Originality:** 2
**Overall Recommendation:** 3
**Confidence:** 4

**Summary:**

This paper introduces BroRL, and RLVR method that increases the number of rollouts generated per prompt significantly. A formal analysis shows how BroRL improves exploration by mitigating the negative affects of so-called "unsampled" tokens. Empirically, BroRL increases performance on standard benchmarks in less training time than vanilla RLVR.

**Compliance With Llm Reviewing Policy:**

Affirmed.

**Key Questions For Authors:**

1. Can you clarify the central contribution of this work? If it's an algorithmic contribution, it's important to compare to other methods. It's a formal hyper parameter analysis, it's important to compare to prior analyses.

**Limitations:**

Limitations are not discussed explicitly. I think the main one is the cost of generating more rollouts.  It leads to better performance, but increasing training cost a lot. Is this approach practical if someone only has small GPUs with low memory?

**Strengths And Weaknesses:**

**Strengths**

The primary contribution (in my view) is the formal analysis in Section 2 characterizing how “unsampled” tokens influence model updates. Unsampled actions are an overlooked aspect of learning from finite data, and the community would benefit from this discussion. I am aware of only one related analysis in the more general RL setting (Corrado et al. [1], I discuss this work more below).  The analysis in this paper is particularly nice because it focuses on quantities relevant to RLHF on LLMs.


**Weaknesses**

1. Many existing works study how the number of rollouts per prompt affects learning (described below). The authors should discuss how these related works compare to this submission. In addition, some of these methods like DARS [4] should serve as empirical baselines or the authors should explain why they are not included. I think some of these works are very recent and therefore do not need to be evaluated against. Having said that, I think the paper would benefit by acknowledging recent preprints as concurrent work in the related work section.
    * Xie et al. [2]  study how hyperparameters such as rollout count affect learning dynamics.
    * Zeng et al. [3 show that increasing rollouts improves reward estimation and reduces policy gradient variance.
    * DARS [4] and AR3PO [5] adaptively allocate more rollouts to difficult problems.
    * EFRame [6] performs additional sampling rounds on hard prompts using higher temperature.
    * Several works [7, 8, 9] argue that rollout diversity can matter more than the number of rollouts.
Note that Xie et al. [2], Yang et al. [4] report that increasing rollout count does not always improve performance. It would be helpful to discuss this observation when motivating why it's a good idea to increase the number of rollouts.

	2.	The method this paper introduces---increasing the number of rollouts generated per response---is ultimately a hyper parameter adjustment. Its simplicity is not a weakness in my view, but this approach does not technically address the issue of unsampled actions. I will explain what I mean more precisely. In the limit of infinite data, there are no unsampled tokens. Practically, we can only train on a finite set of rollouts, so we will almost surely have unsampled tokens. Thus, we have two general ways to address this challenge: either design a method that mitigates the effect of unsampled tokens under finite data, or collect substantially more data so that unsampled tokens become less liekly (which sidesteps the challenge). This paper takes second approach.  This is a fair direction, but it makes the paper’s positioning important. It makes me view that the main contribution is an analysis of how rollout count affects learning. If that is the intended contribution, the paper would benefit from framing it more explicitly as such and compare to related analyses, particularly Xie et al. [2]. If the authors intend it to be interpreted this way, it would strengthen the work to evaluate the method against existing baselines.

**Other comments**

Corrado & Hanna [1] would be a useful addition to the related work. They show how the distribution of finite sample set trajectories will almost surely deviate from the expected distribution. They call this deviation sampling error and then show how sampling error leads to inaccurate policy gradient estimates and potentially cause suboptimal convergence. The concept of sampling error introduced in this work is analogous to the concept of "unsampled tokens". However, the term “unsampled” does not fully capture the phenomenon. Sampling error refers to a mismatch between the empirical and expected distributions, not merely the absence of certain actions/tokens.

**References**

1. Corrado & Hanna. On-Policy Policy Gradient Reinforcement Learning Without On-Policy Sampling. https://arxiv.org/abs/2311.08290.
2. Xie et. al. Demystifying Design Choices of Reinforcement Fine-tuning: A Batched Contextual Bandit Learning Perspective. https://arxiv.org/abs/2601.22532
3. Zeng et al. Shrinking the Variance: Shrinkage Baselines for Reinforcement Learning with Verifiable Rewards. https://arxiv.org/abs/2511.03710
4. Yang et al. Depth-Breadth Synergy in RLVR: Unlocking LLM Reasoning Gains with Adaptive Exploration. https://arxiv.org/abs/2508.13755
5. Zhang et al. Improving Sampling Efficiency in RLVR through Adaptive Rollout and Response Reuse. https://arxiv.org/abs/2509.25808
6. Wang et al. EFRame: Deeper Reasoning via Exploration-Filter-Replay Reinforcement Learning Framework. https://arxiv.org/pdf/2506.22200
7. Xu et al. Not All Rollouts Are Useful: Down-Sampling Rollouts in LLM Reinforcement Learning. https://arxiv.org/abs/2504.13818
8. Bai et al. M-GRPO: Stabilizing Self-Supervised Reinforcement Learning for Large Language Models with Momentum-Anchored Policy Optimization. https://arxiv.org/abs/2512.13070
Emphasizes maintaining diverse rollouts to stabilize policy optimization and avoid collapse during self-supervised RL training.
9. Cui et al. The Entropy Mechanism of Reinforcement Learning for Reasoning Language Models. https://arxiv.org/abs/2505.22617
Analyzes how entropy and rollout diversity support effective reasoning improvements in RL-trained language models.

---

> ### Author Rebuttal · Authors · 2026-03-29
>
> We thank the reviewer for the detailed feedback. We especially appreciate the positive assessment of the formal analysis in Section 2 and the careful enumeration of related work.
>
> ##  On related work and empirical baselines
>
> We agree the related work section needs expansion and will add all nine cited papers. We position the most relevant ones below.
>
> Corrado & Hanna [1] analyze "sampling error" in general RL: finite on-policy trajectories systematically deviate from the expected distribution, leading to high-variance gradient estimates. Their PROPS method corrects this via adaptive off-policy sampling that increases the probability of under-sampled actions. Our "unsampled tokens" formulation captures an analogous phenomenon in the RLVR/LLM setting. We will acknowledge this connection explicitly. The distinction is that their "sampling error" is a distributional mismatch concept, while our analysis provides a specific decomposition (Theorem 1) of how this mismatch affects correct-mass dynamics through the unsampled coupling term. We agree the reviewer's point that "unsampled" does not fully capture the phenomenon, and we will revise the terminology to better reflect this.
>
> Xie et al. [2] study hyperparameter effects including rollout count from a batched contextual bandit perspective. Their analysis is complementary: they characterize when increasing rollouts helps or hurts from an optimization lens; we provide a mass-balance decomposition identifying the specific mechanism (the coupling term decay in Lemma 1).
>
> DARS [4] and AR3PO [5] adaptively allocate rollouts based on prompt difficulty. DARS shows that under a fixed compute budget, uniform rollout allocation is suboptimal compared to difficulty-adaptive allocation. This is consistent with BroRL's theory and actually motivates a natural extension: the unsampled coupling effect from our Theorem 1 is stronger for hard prompts (where correct-token probability mass is smaller and more tokens remain unsampled) than for easy prompts (where most rollouts already produce correct answers). So the theory predicts exactly what DARS finds empirically. Combining BroRL's large-N baseline with difficulty-adaptive allocation is a natural next step (see our response to Reviewer 98hh Q3 for a concrete adaptive rule based on our positivity margin M(N)). We view DARS/AR3PO as complementary rather than competing approaches.
>
> Regarding DARS as a baseline: DARS uses different base models (Qwen2.5-Math), different training setups, and targets a different metric. A controlled comparison would require re-implementing DARS in our framework with the same model and compute budget. We acknowledge this as valuable future work.
>
> EFRame [6] adds sampling rounds at higher temperature for hard prompts. Works [7,8,9] emphasize rollout diversity. BroRL provides a distinct theoretical explanation for why increasing N helps (coupling term decay per Lemma 1), while these works offer complementary exploration strategies. We will discuss all of them.
>
> ##  On central contribution and positioning
>
> The reviewer correctly identifies that increasing N sidesteps rather than directly solves the finite-sample problem. Our Theorem 1 shows exactly this: as N grows, U_pos,2 and U_neg,2 shrink, making the coupling term vanish and the update reliably positive. The contribution is: (1) a formal analysis of how finite rollout groups create learning instabilities through unsampled-token coupling, providing a mechanism-level explanation for training stalls; (2) an empirical finding that increasing rollout breadth revives training after saturation (62% vs 41% dynamic sampling pass rate, 72.4 vs 36.5 samples/s throughput); and (3) a practical recipe. We will revise the framing to position this explicitly as a theoretically grounded analysis of rollout-count effects on learning, and compare more directly to Xie et al. [2].
>
> ##  On cost and practicality
>
> BroRL requires generating 512 rollouts per prompt. In our setup, this is partially offset by hardware efficiency: the shift from memory-bound to compute-bound generation nearly doubles throughput (72.4 vs 36.5 samples/s). For practitioners with limited GPUs, the ProRL-then-BroRL recipe accommodates constraints naturally: use smaller N early, scale up when plateaus are reached and hardware supports it. We will add explicit discussion of this tradeoff.

---

> > ### Author Rebuttal · Reviewer_BUL1 · 2026-04-03
> >
> > Thank you for your response. I'm mostly satisfied but have follow-up comments.
> >
> > First, I want to say that repositioning the paper to emphasize the contribution as a formal analysis of how the number of rollouts sampled affects learning is most important to me. The authors should also incorporate discussion on how BroRL relates to other rollout sampling methods and how BroRL relates to "sampling error" to properly position this work in the literature.
> >
> > After ruminating on this paper for a while, I'd like to rephrase my comment about comparing this work to existing methods. The author's are correct; method like DAPO, DARS, etc. are complementary to BroRL.  Having said that, I think the community would appreciate this work more if additional experiments show that  BroRL tacks upon existing methods. In particular, you can increase the number of rollouts for most RL algorithms, including DAPO and DARS. If BroRL does not improvement performance on top of these methods (which presumably require less compute as they are typically run with relatively few rollouts), practitioners would prefer running DAPO/DARS without BroRL.

---

### Official Review · Reviewer_6Qew · 2026-03-19

**Soundness:** 3
**Presentation:** 3
**Significance:** 3
**Originality:** 3
**Overall Recommendation:** 4
**Confidence:** 4

**Summary:**

The paper proposes BroRL, which is said to scale RLVR by increasing the number of rollouts per prompt N rather than training for more steps, to escape from performance plateaus, yielding continuous performance gains beyond the saturation point observed in ProRL when scaling the number of training steps. They prove that the first two terms are always non-negative and that the coupling term shrinks as N grows. Empirically, BroRL (N=512, B=128) trained on a 1.5B model achieves 63.03 on Math, 54.20 on Code, and 63.09 on Reasoning Gym benchmarks, compared to ProRL's (N=16, B=512) plateau and degradation. Additional experiments on a 4B model confirm the pattern. The approach also shows improved hardware utilization, shifting from memory-bound to compute-bound operation.

**Compliance With Llm Reviewing Policy:**

Affirmed.

**Key Questions For Authors:**

- The experiments start from a saturated ProRLv2 checkpoint. Would BroRL show similar advantages when training from a freshly fine-tuned model, or is the benefit primarily about escaping plateaus that ProRL gets stuck in?
- The experimental comparison focuses on ProRL, but does not include other recent RL scaling recipes such as JustRL [1] and QuestA [2]. JustRL achieves competitive performance with significantly less compute, and QuestA simplifies the RL pipeline without requiring multiple stages, both achieving strong baselines.

[1] JustRL: Scaling a 1.5B LLM with a Simple RL Recipe

[2] QuestA: Expanding Reasoning Capacity in LLMs via Question Augmentation

**Limitations:**

Yes.

**Strengths And Weaknesses:**

### Strengths

- The experimental setup is reasonably careful. The matched wall-clock comparison between BroRL and ProRL is the right comparison to make.
- The method is simple and easy to follow, with empirical gains and a theoretical foundation.
- The paper is well-structured and well-written. And the practical finding is good: the simple recipe of "use more rollouts, fewer prompts" under a fixed compute budget would be immediately useful for practitioners.


### Weaknesses
- While providing the theoretical framing, the practical insight reduces to a hyperparameter scaling recommendation. The method is just setting N to 512 instead of 16, which is interesting but is quite focused. The current contribution feels more like a useful empirical observation (perhaps suitable for a blog or technical report) than a broadly applicable scientific finding.
- In Table 2, the Math Score improves from 61.69 (baseline) to 62.85 (BroRL +134 steps) vs 62.02 (ProRL +535 steps). The difference of 0.8 is modest. Whether this gain justifies the additional cost is worth investigating.

---

> ### Author Rebuttal · Authors · 2026-03-29
>
> We thank the reviewer for the careful reading and constructive feedback.
>
> ## On the contribution being a hyperparameter recommendation
>
> We agree BroRL is simple, and we view that as a strength. The contribution has three parts: (1) a formal analysis showing that unsampled tokens create a coupling term in the RLVR update that can be negative, and that this term shrinks as N grows (Theorem 1, Lemma 1); (2) an empirical finding that scaling rollout breadth breaks through plateaus where step-scaling stagnates and regresses; and (3) a practical recipe that integrates into existing RLVR pipelines without architectural changes. We will revise the framing to make clear that BroRL is a theoretically motivated scaling recipe for the post-saturation regime, not a new RL algorithm.
>
> ##  On the modest gain in Table 2
>
> The key comparison is trajectory direction, not absolute magnitude. After saturation, ProRL with 535 additional steps (133.8h) regresses: Math 62.08 to 62.02, Reasoning Gym 62.10 to 61.45. BroRL at 122.8h reaches 62.85/53.48/62.82 and keeps climbing to 63.03/54.20/63.09 by 173.8h, with no sign of regression on any benchmark. The practical question is: after saturation, does BroRL yield a better return on additional compute than continuing ProRL? Our data says yes across all three domains.
>
> ##  On whether BroRL is primarily about escaping plateaus
>
> Yes, and we consider this the correct framing. Our recommendation is: use ProRL-style training (small N, many prompts) for fast early convergence, then switch to BroRL (large N) when progress stalls. We will revise to present BroRL as a late-stage scaling strategy.
>
> ##  On comparison with JustRL and QuestA
>
> We will discuss both in the revision.
>
> JustRL uses N=8 rollouts with 16K context on 32 A800 GPUs, trained from a distilled checkpoint (DeepSeek-R1-Distill-Qwen-1.5B). It achieves 54.87% average on math benchmarks vs ProRL-V2's 53.08% using a single-stage recipe. Direct comparison with BroRL is not straightforward: JustRL starts from a distilled checkpoint and trains from scratch, while BroRL continues training from a ProRL checkpoint that has already undergone 3K RL steps. The two target different training regimes. JustRL's 16K context also incurs higher per-sample generation cost than our 8K-context baseline, making cross-setting compute comparison nontrivial.
>
> QuestA augments hard problems with partial solutions as hints, uses 32K context, and trains on OpenMath-Nemotron-1.5B. This is a data augmentation approach complementary to BroRL's rollout breadth scaling; the two could potentially be combined.
>
> These concurrent works share a common theme: exploration quality matters in RLVR. JustRL finds that adding "standard tricks" can collapse exploration. DARS (Yang et al.) shows that under a fixed compute budget, uniformly allocating rollouts across all prompts is less efficient than adaptive allocation that concentrates rollouts on harder problems. BroRL provides a theoretical mechanism (the unsampled coupling term in Theorem 1) explaining why broader exploration helps in the first place, and validates it in the post-saturation regime. DARS and BroRL are complementary: BroRL explains why more rollouts help; DARS shows where to allocate them most efficiently. Combining broad N scaling with difficulty-adaptive allocation is a natural next step (see our response to Reviewer 98hh Q3 for details).

---

> > ### Author Rebuttal · Reviewer_6Qew · 2026-04-04
> >
> > Thanks for the rebuttal. I think the  theoretical motivation and empirical results are sound.

---

### Decision · Program_Chairs · 2026-04-30

**Decision:**

Accept (regular)

**Comment:**

The paper presents a different view on rollouts in RL, advocating broader rollout with more concurrent sampling, arguing it prevents performance plateaus effectively. Reviewers note the good experiments, the simplicity of the method and interesting theoretical analysis. However, the limited contribution, i.e. setting one hyperparameter differently, some trivial aspects in the theoretical analysis, and limited empirical comparisons are criticized.

This work is a borderline paper.